# Cardiac neurons expressing a glucagon-like receptor mediate cardiac arrhythmia induced by high-fat diet in *Drosophila*

Yunpo Zhao[1,2], Jianli Duan[1,2], Joyce van de Leemput[1,2], Zhe Han[1,2]*

[1]Center for Precision Disease Modeling, Department of Medicine, University of Maryland School of Medicine, Baltimore, United States; [2]Division of Endocrinology, Diabetes and Nutrition, Department of Medicine, University of Maryland School of Medicine, Baltimore, United States

*For correspondence:
zhan@som.umaryland.edu

Competing interest: The authors declare that no competing interests exist.

## eLife Assessment

This study reports **useful** information on the mechanisms by which a high-fat diet induces arrhythmias in the model organism *Drosophila*. Specifically, the authors propose that adipokinetic hormone (Akh) secretion is increased with this diet, and through binding of Akh to its receptor on cardiac neurons, arrhythmia is induced. The authors have revised their manuscript, but in some areas, the evidence remains **incomplete**, which the authors say future studies will be directed to closing the present gaps. Nonetheless, the data presented will be helpful to those who wish to extend the research to a more complex model system, such as the mouse.

**Abstract** Cardiac arrhythmia leads to increased risks for stroke, heart failure, and cardiac arrest. Arrhythmic pathology is often rooted in the cardiac conduction system, but the mechanism is complex and not fully understood. For example, how metabolic diseases, like obesity and diabetes, increase the risk for cardiac arrhythmia. Glucagon regulates glucose production, mobilizes lipids from the fat body, and affects cardiac rate and rhythm, attributes of a likely key player. *Drosophila* is an established model to study metabolic diseases and cardiac arrhythmias. Since glucagon signaling is highly conserved, we used high-fat diet (HFD)-fed flies to study its effect on heart function. HFD led to increased heartbeat and an irregular rhythm. The HFD-fed flies showed increased levels of adipokinetic hormone (Akh), the functional equivalent to human glucagon. Both genetic reduction of Akh and eliminating the Akh-producing cells (APC) rescued HFD-induced arrhythmia, whereas heart rhythm was normal in Akh receptor mutants (*AkhR^null^*). Furthermore, we discovered a pair of cardiac neurons that express high levels of Akh receptor. These are located near the posterior heart, make synaptic connections at the heart muscle, and regulate heart rhythm. Altogether, this Akh signaling pathway provides new understanding of the regulatory mechanisms between metabolic disease and cardiac arrhythmia.

## Introduction

Arrhythmia refers to an irregular, decreased (bradycardia), or increased (tachycardia) heartbeat. Temporary disruption is usually benign; however, chronic arrhythmia has been linked to significantly increased risks for stroke, heart failure, and cardiac arrest (*Kannel et al., 1998*; *Kannel et al., 1983*; *Nattel et al., 2014*; *Roberts-Thomson et al., 2011*). Its pathogenesis is rooted in the cardiac conduction system; however, the mechanism is complex and much remains unknown. Two well-established risk factors that directly contribute to the development of cardiovascular disorders and arrhythmia

are obesity (*Gupta et al., 2022*; *Powell-Wiley et al., 2021*) and diabetes mellitus (*Aune et al., 2018*; *Huxley et al., 2011*; *Lee et al., 2017*). In fact, a longitudinal study into obesity (13.7 years mean follow-up; 5282 participants) found a 4% increased risk for atrial fibrillation (a form of arrhythmia) per one-unit increased body mass index (*Wang et al., 2004*). The other major risk factor, diabetes mellitus, has been shown to impact the cardiac conduction system, leading to increased risk of developing atrial fibrillation and ventricular arrhythmias (*Kannel et al., 1998*). A meta-analysis found that patients with diabetes had a 28–40% increased risk for developing atrial fibrillation, with a 20% risk increase reported for pre-diabetic patients (*Aune et al., 2018*; *Huxley et al., 2011*). The relationship between higher blood glucose levels and increased risk for atrial fibrillation was dose-responsive (*Aune et al., 2018*). Diabetes mellitus, when considered a cause of disrupted metabolism, as well as obesity, has been independently associated with increased risk for atrial fibrillation (*Lee et al., 2017*). However, to what extent diabetes, blood glucose, and obesity contribute to atrial fibrillation, independently or collectively, and through which pathomechanism requires further study.

Antagonistic actions by glucagon and insulin regulate glucose metabolism. Besides regulating glucose release from the liver, glucagon facilitates the release of glucose as well as lipids from the fat body, acts as a satiety factor in the central nervous system, affects the glomerular filtration rate, and regulates intra-islet secretion of insulin, glucagon, and somatostatin to meet increased energy demands (*Habegger et al., 2010*; *Heppner et al., 2010*; *Vuguin and Charron, 2011*). These glucagon regulatory effects are evident in patient studies that showed that an impaired counter-regulatory glucagon response, observed as increased free plasma insulin levels, contributes to glucose instability in patients with long-term diabetes *Scott et al., 1980*; that consumption of dietary fats leads to increased plasma glucagon levels in healthy volunteers *Radulescu et al., 2010*; and that plasma glucagon levels were significantly higher in people considered obese compared to those considered lean (*Stern et al., 2019*). Glucagon has been repeatedly shown to affect heart contraction and heart rate. However, the nature of this effect is complex; whether glucagon acts anti- or pro-arrhythmogenic seems to depend on context, such as a non-failing heart or a heart at acute or chronic failure (*Neumann et al., 2023*). That said, glucagon-producing tumors, that is, glucagonomas, can cause tachycardia (a form of arrhythmia defined as >100 heart beats per minute) and heart failure without secondary cause (*Chang-Chretien et al., 2004*; *Zhang et al., 2014*). Moreover, glucagonoma tumor resection resulted in a normalized heart rate and a return to typical heart size and function (*Chang-Chretien et al., 2004*). Similarly, glucagon infusion in healthy human volunteers induced arrhythmias (*Jaca et al., 2002*; *Markiewicz et al., 1978*). Thus, while glucagon clearly affects heart rhythm, when considering the underlying mechanism, much remains unknown.

The link between high-fat diet (HFD), glucagon, and cardiac arrhythmia is conserved and well-established in animal models. In fact, studies from the 1960s already showed that glucagon increases heart rate in a variety of mammalian species, including dogs, cats, guinea pigs, and rats (*Farah and Tuttle, 1960*; *Lucchesi et al., 1968*). More recently, it was shown in mice that both glucagon and the glucagon receptor (Gcgr) are involved in heart rate regulation (*Mukharji et al., 2013*; *Sowden et al., 2007*). *Drosophila* adipokinetic hormone (Akh), the functional equivalent to human glucagon, is expressed in a small cluster of endocrine neurons, Akh-producing cells (APC) (*Kim and Rulifson, 2004*; *Lee and Park, 2004*). These cells in the corpora cardiaca near the esophagus function similarly to islet cells in mammals, including the mechanism that regulates hormone secretion, and are essential for larval glucose homeostasis (*Kim and Rulifson, 2004*). Like glucagon, Akh is known to also mobilize lipids from the fat body to regulate glucose levels, in the form of trehalose, in the circulating hemolymph to accommodate increased energy demand (*Isabel et al., 2005*). Likewise, increased Akh increases the heart rate in flies (*Noyes et al., 1995*). These studies together suggest that glucagon and glucagon signaling, aside from regulating blood sugar levels, play an evolutionarily conserved role in heart rate regulation.

*Drosophila melanogaster* is a well-established model to study heartbeat and heart arrhythmia (*Birse et al., 2010*; *Ding et al., 2022*; *Ocorr et al., 2007*). For example, HFD-fed flies were used to study the genetic mechanisms of cardiac dysfunction in obesity. It found that HFD leads to reduced cardiac contractility and a reduced heart period (*Birse et al., 2010*). Notably, this phenotype was attenuated by intervention at the insulin-TOR signaling pathway (*Birse et al., 2010*), thus supporting a connection between obesity, glucagon, insulin, and cardiac function. Here, we used *Drosophila* fed a HFD to study heart arrhythmia. HFD led to increased Akh in the APC of the corpora cardiaca. We then

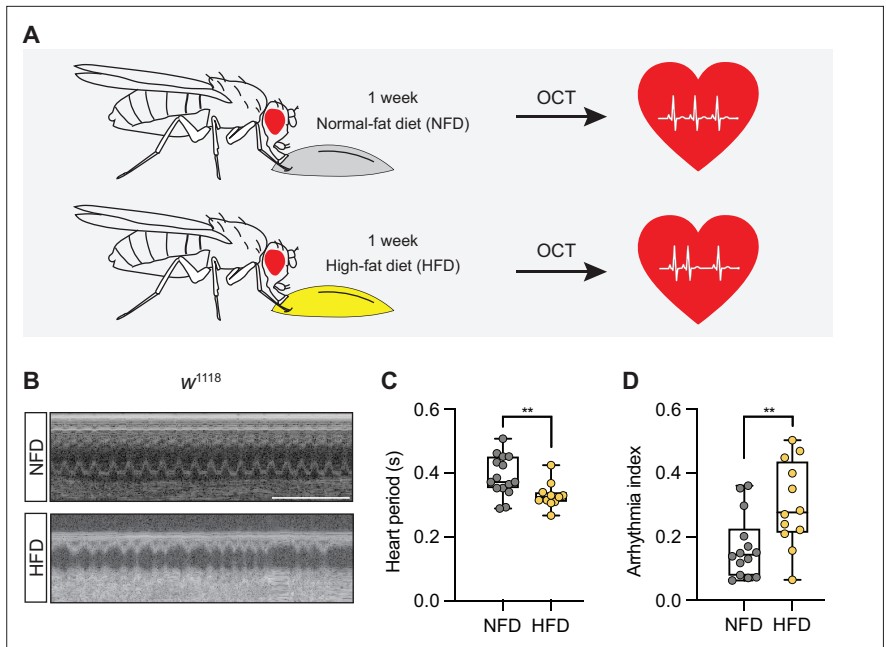

**Figure 1.** High-fat diet (HFD) increases heart rate and arrhythmia index in *Drosophila*. (**A**) Schematic illustration of the experimental design. Adult flies (female) were fed either a normal fat diet (NFD) or an HFD for 7 days following eclosion from pupa, then subjected to heart functional analysis using optical coherence tomography (OCT) to determine the heart period and the arrhythmia index. (**B**) Representative images obtained from OCT videos of $w^{1118}$ flies that were fed NFD or HFD as indicated. Scale bars: 2 seconds. (**C, D**) Quantitation of heart period (**C**; n=15 NFD, n=11 HFD) and arrhythmia index (**D**; n=14 NFD, n=12 HFD). Statistical analysis was performed using *t*-test corrected with Welch; **p<0.01. The numerical data used to generate the figure are provided in *Figure 1— source data 1*.

The online version of this article includes the following source data for figure 1:

**Source data 1.** The numerical data used to generate the *Figure 1*.

identified a hitherto undescribed pair of cardiac neurons near the posterior heart that highly express the Akh receptor (AkhR) and directly innervate the fly heart. We show that these AkhR cardiac neurons (ACN) regulate heart rhythm in the flies and mediate the HFD-induced arrhythmia. Given the conservation of Akh/glucagon signaling, these findings likely have implications for arrhythmia in patients. Given the significance of the vagus nerve in cardiac rhythm (*Ambache and Lippold, 1949*; *Cai et al., 2023*; *Freeman, 1951*; *González et al., 2023*; *Kharbanda et al., 2023*; *van Weperen and Vaseghi, 2023*), it could fulfill a similar function.

## Results

### High-fat diet increases heart rate and arrhythmia index

All flies ($w^{1118}$) that eclosed (i.e., adults that emerged from pupa) within 8 hours were sorted (females only) and transferred to fresh food vials (25 flies per vial) containing normal fat diet (NFD) or HFD (NutriFly diet supplemented with 14% fat) for 7 days. At which point, we determined the heartbeat of the flies using optical coherence tomography (OCT) (*Figure 1A*). Flies on a HFD show a significantly reduced heart period and increased arrhythmia index compared to NFD-fed flies (*Figure 1B–D*). These findings are in agreement with previous work (*Birse et al., 2010*) and indicate that HFD imposes a pathogenic effect on heart function.

### High-fat diet up-regulates Akh expression and increases Akh-producing cell (APC) activity

The ingestion of superfluous macronutrients has a great impact on renal function, feeding behavior, and metabolism (*Huang et al., 2020*; *Liao et al., 2021*; *Zhao et al., 2025*; *Zhao et al., 2023*; *Zhao*

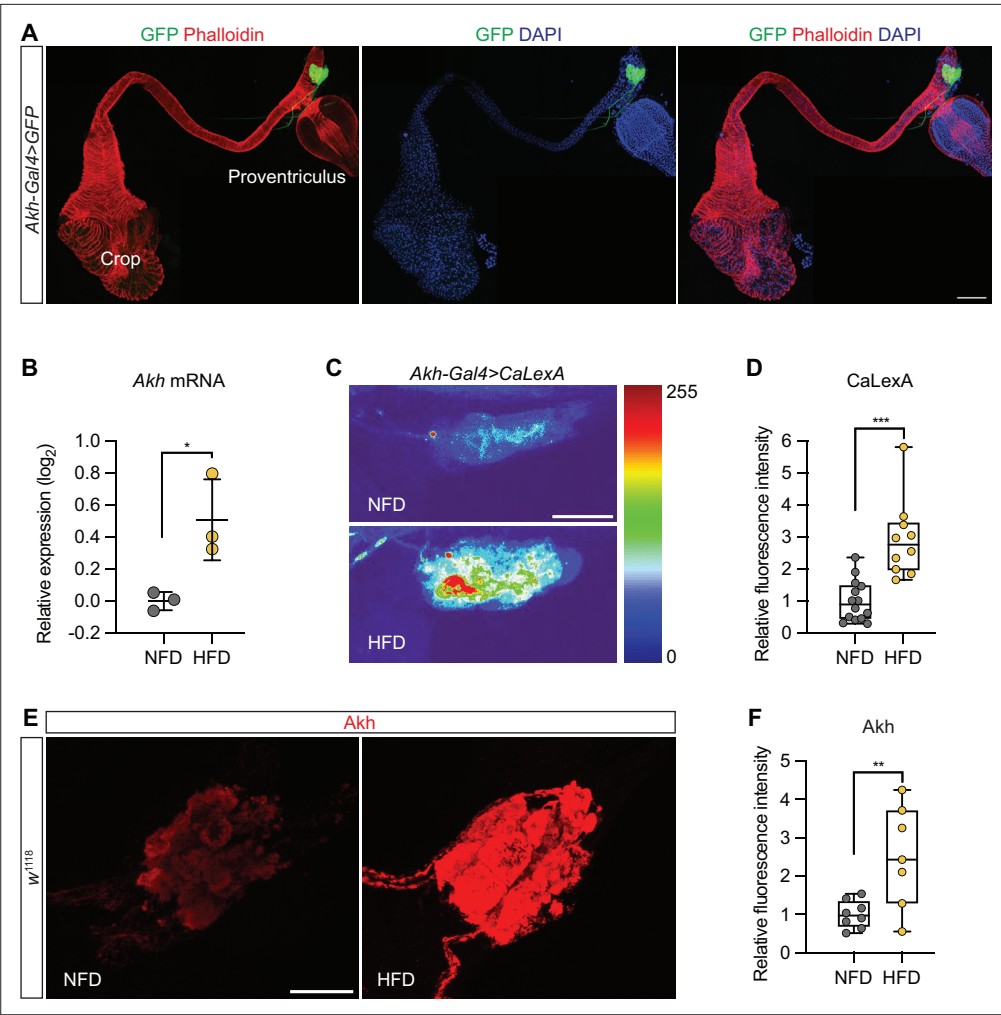

**Figure 2.** High-fat diet (HFD) upregulates adipokinetic hormone gene *Akh* and activates Akh neurons. (**A**) Representative confocal image of Akh-producing cells (APCs; green, GFP) in *Akh*-Gal4>10xUAS-*GFP* (*Akh*-Gal4>*GFP*) transgenic flies (3-day-old females). The image shows the anterior section of the digestive system, including the crop, esophagus, and proventriculus. The APCs are located at the corpora cardiaca, which attaches to the esophagus, just anterior to the proventriculus. Phalloidin stains actin filaments red; DAPI stains DNA in nucleus blue. Scale bar: 100 μm. (**B**) RT-qPCR analysis of *Akh* expression in $w^{1118}$ flies (female) that were fed either a normal fat diet (NFD) or a high-fat diet (HFD) for 7 days following eclosion from pupa. Ten flies per group, repeated three times. Statistical analysis was performed with unpaired *t*-test corrected with Welch; *p<0.05. Error bars represent SD. (**C**) Representative confocal images of *Akh*-Gal4>*CaLexA* APCs from female adult flies fed an NFD or HFD (for 7 days following eclosion from pupa). CaLexA is a transcription-based genetically encoded calcium indicator for neuronal activity. CaLexA fluorescence has been presented as a heatmap; scale from 0 (blue), no CaLexA detected, to 255 (red) high levels CaLexA. Scale bar: 20 μm. (**D**) Quantitation of CaLexA fluorescence in C (7-day-old females; NFD, n=14; HFD, n=10). Statistical analysis was performed with unpaired *t*-test corrected with Welch; ***p<0.001. (**E**) Representative confocal images of $w^{1118}$ APCs from female adult flies fed an NFD or HFD (for 7 days following eclosion from pupa). Anti-Akh is in red. DAPI stains DNA in blue. Scale bar: 20 μm. (**F**) Quantitation of anti-Akh fluorescence in E (7-day-old females; NFD, n=8; HFD, n=7). Statistical analysis was performed with unpaired t-test corrected with Welch; **p<0.01. The numerical data used to generate the figure are provided in *Figure 2—source data 1*.

The online version of this article includes the following source data and figure supplement(s) for figure 2:

**Source data 1.** The numerical data used to generate the *Figure 2*.

**Figure supplement 1.** High-fat diet (HFD) increases crop size.

**Figure 3.** Akh regulates the heartbeat. (**A**) Representative images obtained from optical coherence tomography (OCT) videos of control (*Akh*-Gal4) and *Akh*-RNAi (*Akh*-Gal4>UAS-*Akh*-RNAi) flies (females, 7 days old) that were fed a normal fat diet (NFD) or a high-fat diet (HFD), as indicated, for seven days starting at eclosion from pupa. Scale bars: 2 seconds. (**B**) Quantitation of the arrhythmia index in control (n=15 NFD, n=13 HFD) and *Akh*-RNAi (n=13 NFD, n=14 HFD) flies. Statistical analysis was performed with two-way ANOVA corrected with Sidak. Statistical significance: *p<0.05; ****p<0.0001; ns, not significant. The numerical data used to generate the figure are provided in *Figure 3—source data 1*.

The online version of this article includes the following source data and figure supplement(s) for figure 3:

**Source data 1.** The numerical data used to generate the *Figure 3*.

**Figure supplement 1.** *Akh*-Gal4>*Akh* IR-depleted Akh.

*et al., 2022*). We observed enlarged crops, the functional equivalent to the human stomach, in the HFD-fed flies (*Figure 2—figure supplement 1*). We asked whether the metabolic signaling pathways contributed to the HFD-caused pathogenic effect on the heart. Akh, the functional equivalent to human glucagon, is expressed by a group of neuroendocrine cells known as APC in the corpora cardiaca (*Lee and Park, 2004*). Indeed, flies that carried *Akh*-Gal4 to express GFP showed Akh-producing cells at the corpora cardiaca, which attaches to the esophagus, just anterior to the proventriculus (*Figure 2A*). Inconsistent with a previous work (*Liao et al., 2021*), we showed that the expression of *Akh* was significantly up-regulated in the flies fed a HFD, compared to NFD-fed flies (*Figure 2B*).

Next, to test whether HFD affects APC neuron activity, we used CaLexA (*Masuyama et al., 2012*). A basal level of CaLexA fluorescence was observed in the APC of NFD-fed flies, while a significantly higher fluorescence was detected in the APC of HFD-fed flies (*Figure 2C and D*). This shows that HFD increases APC neuron activity. Taken together, our data show that HFD activates Akh expression in the APC and increases APC activity.

## Akh regulates heartbeat

To confirm the importance of APC activity and Akh release by the APC, we downregulated *Akh* (*Akh*-Gal4>UAS-*Akh*-RNAi) and analyzed its effect on heart function. Immunostaining showed diminished anti-Akh (*Lee and Park, 2004*) fluorescence (*Figure 3—figure supplement 1*), indicating the RNAi efficiency. As expected, upon Akh depletion in the APCs, the difference in arrhythmia between the NFD and HFD-fed flies disappeared (*Figure 3A and B*). The findings indicate that the endocrine signal Akh, originating in the APC neurons, mediates the HFD cardiac functional pathogenicity.

## AkhR mediates the high-fat diet pathogenic effect on the heartbeat

Akh binds its receptor AkhR, a G-protein coupled receptor, to activate the signaling pathway (*Staubli et al., 2002*). We quantified *AkhR* expression using RT-qPCR and observed significantly up-regulated expression in HFD-fed flies (*Figure 4A*). To confirm its role in Akh-mediated HFD-induced arrhythmia, we tested *AkhR* null mutant flies (*AkhR^null^*). In line with Akh depletion, flies with *AkhR* mutation and fed a HFD did not show the HFD-associated cardiac arrhythmia (*Figure 4B and C*). These data show that Akh/AkhR signaling mediates the pathogenic effect of HFD on heart function.

## A pair of AkhR cardiac neurons (ACN) are associated with the heart

To determine the AkhR expression pattern, we used *AkhR*-Gal4 (*Lee et al., 2018*) to drive the expression of GFP. The adipose fat body tissue is a major organ that expresses AkhR at the embryonic and larval stages (*Grönke et al., 2007*), as well as in the adult (*Bharucha et al., 2008*). Likewise, we

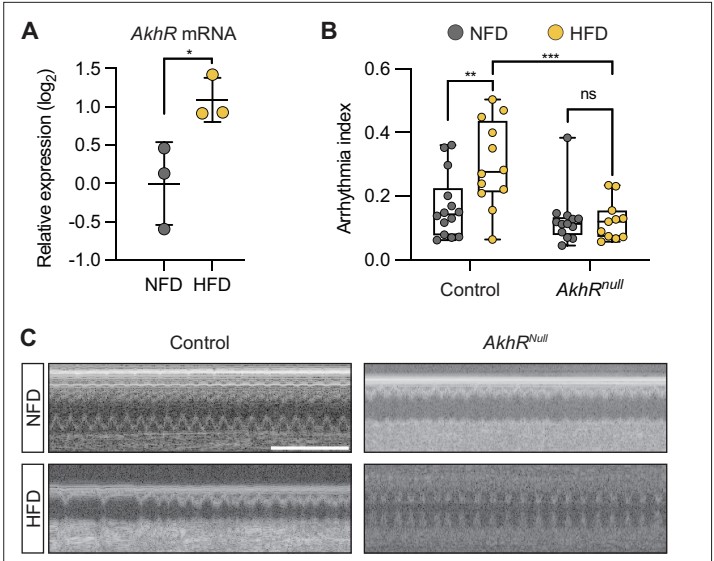

**Figure 4.** AkhR regulates the heartbeat. (**A**) RT-qPCR analysis of *AkhR* expression in *w*[1118] flies (female) that were fed either a normal fat diet (NFD) or a high-fat diet (HFD) for 7 days following eclosion from pupa. Ten flies per group, repeated three times. Statistical analysis was performed with unpaired *t*-test corrected with Welch; *p<0.05. Error bars represent SD. (**B**) Quantitation of arrhythmia index in control (n=14 NFD, n=12 HFD) and *AkhR*[null] (n=13 NFD, n=11 HFD) flies. Statistical analyses were performed with two-way ANOVA corrected with Sidak; **p<0.01; ***p<0.001; ns, not significant. (**C**) Representative images obtained from optical coherence tomography (OCT) videos of control (*w*[1118]) and *AkhR*[null] mutant flies (females, 7 days old) that were fed NFD or HFD, as indicated, for seven days starting at eclosion from pupa. Scale bars: 2 seconds. The numerical data used to generate the figure are provided in *Figure 4—source data 1*.

The online version of this article includes the following source data for figure 4:

**Source data 1.** The numerical data used to generate the *Figure 4*.

observed GFP fluorescence in the fat body in *AkhR*-Gal4>*GFP* flies (*Figure 5A and B*). Notably, no fluorescence was detected in the cardiac muscles. However, we did find two neurons with strong GFP fluorescence, indicative of high expression levels of AkhR, located near the posterior end of the heart tube (*Figure 5B*). These neurons had elaborate neurites along the heart tube (*Figure 5B–D*) and formed synaptic connections with heart muscles, as revealed by immunostaining for active zone marker Bruchpilot (Brp) (*Figure 5E and F*). These two neurons likely communicate with the heart muscle via these neuromuscular junctions. Therefore, we refer to these two neurons as AkhR cardiac neurons (ACN). Immunostaining for Akh showed positive fluorescence on the ACN (*Figure 5—figure supplement 1*), suggesting that the cardiac neurons receive Akh signal.

## Partial elimination of AkhR cardiac neurons (ACN) causes arrhythmia

To determine the function of the ACN, we set out to eliminate the pair. We overexpressed UAS-*rpr* under the control of *AkhR*-Gal4 to induce apoptosis. We observed one remaining AkhR cardiac neuron in the *AkhR*-Gal4>UAS *rpr* flies (*Figure 6A*), indicating partial elimination. The *AkhR*-Gal4>UAS *rpr* flies were subjected to OCT analysis. The profile and rhythm of the heartbeat were drastically affected in the flies with only one ACN (*Figure 6B*). This demonstrates the importance of the ACN in regulating the heartbeat.

## Discussion
### The role of AkhR, glucagon-like receptor, in regulating heart rate and rhythm

Heart function is dependent on ATP continuous synthesis. Cardiac ATP comes from fatty acids, glucose, and lactate (*Kodde et al., 2007*). Glucagon converts stored glycogen, in the liver and fat

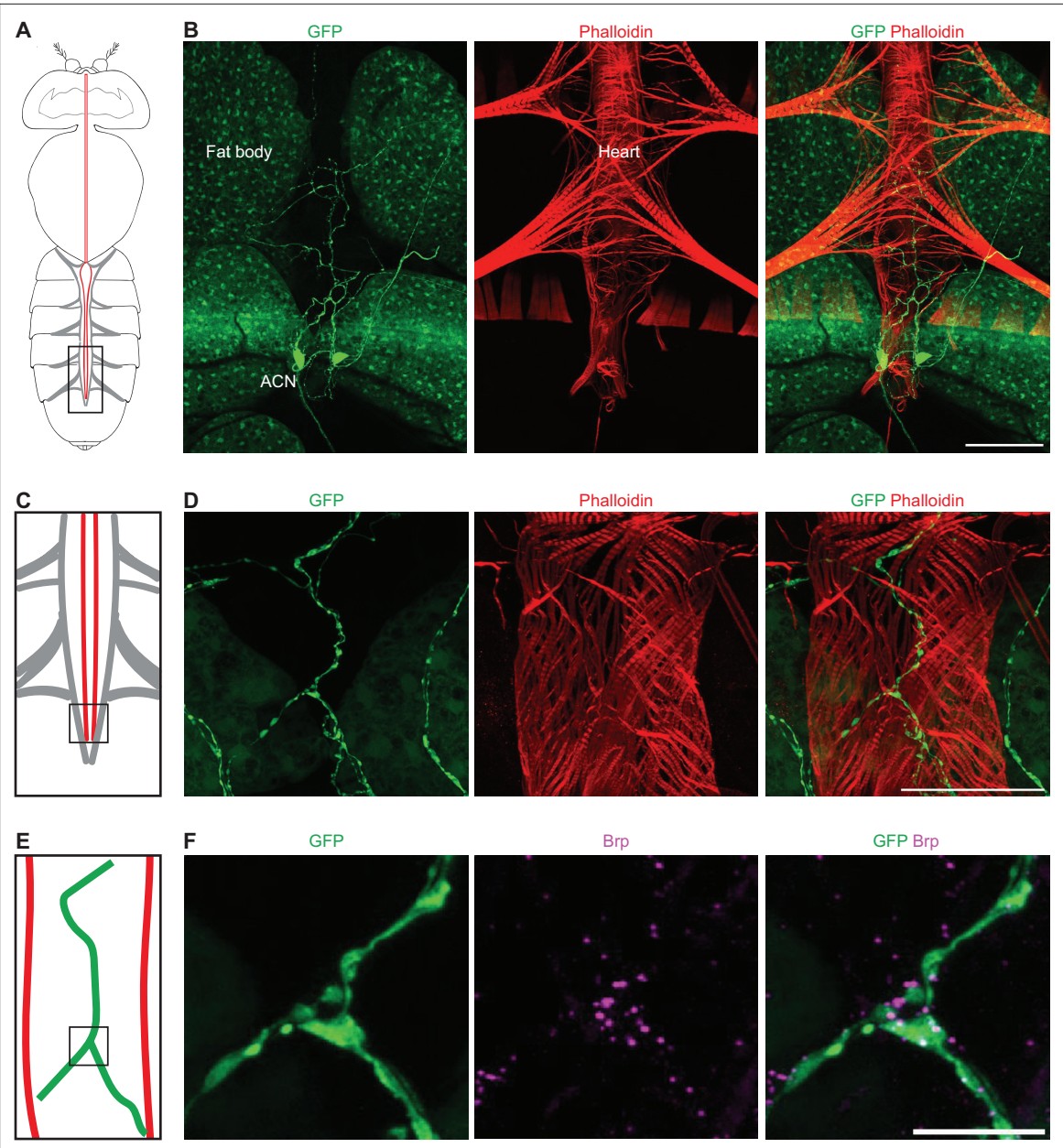

**Figure 5.** AkhR cardiac neurons (ACNs) form synaptic connections with the posterior heart. (**A**) Graphic depiction of the body (no legs or wings) of an adult fly with the head oriented toward the top. The heart tube (red) is located along the midline in the abdomen (bottom of graphic), with the alary muscles that connect the heart to the exoskeleton represented in light gray. (**B**) A representative confocal image of an *AkhR*-Gal4>10xUAS-*GFP* adult fly (7-day-old, female). The dorsal region of an abdomen corresponding to the boxed region in (**A**) is shown. AkhR >*GFP* labels AkhR in green. ACN, AkhR cardiac neuron. Phalloidin stains actin filaments red. Scale bar: 100 μm. (**C**) Schematic illustration of the posterior fly heart. Corresponds to boxed region in (**A**). (**D**) A representative confocal image of the posterior heart of an *AkhR*-Gal4>10xUAS-*GFP* fly (7-day-old, female) corresponding to the boxed region in (**C**). AkhR>*GFP* labels AkhR in green. Phalloidin stains actin filaments red. Scale bar: 50 μm. (**E**) Schematic illustration of the posterior end of an adult fly heart. The neurites of the cardiac neurons are depicted in green. Relates to the boxed region in (**C**). (**F**) Confocal image of the posterior heart of an *AkhR*-Gal4>10xUAS-*GFP* fly (7-day-old, female) corresponding to the boxed region in (**E**). AkhR>*GFP* labels AkhR in green. Anti-Bruchpilot (Brp), a marker for the active zone, is shown in magenta. Scale bar: 10 μm.

The online version of this article includes the following figure supplement(s) for figure 5:

**Figure supplement 1.** AkhR neuron receives Akh.

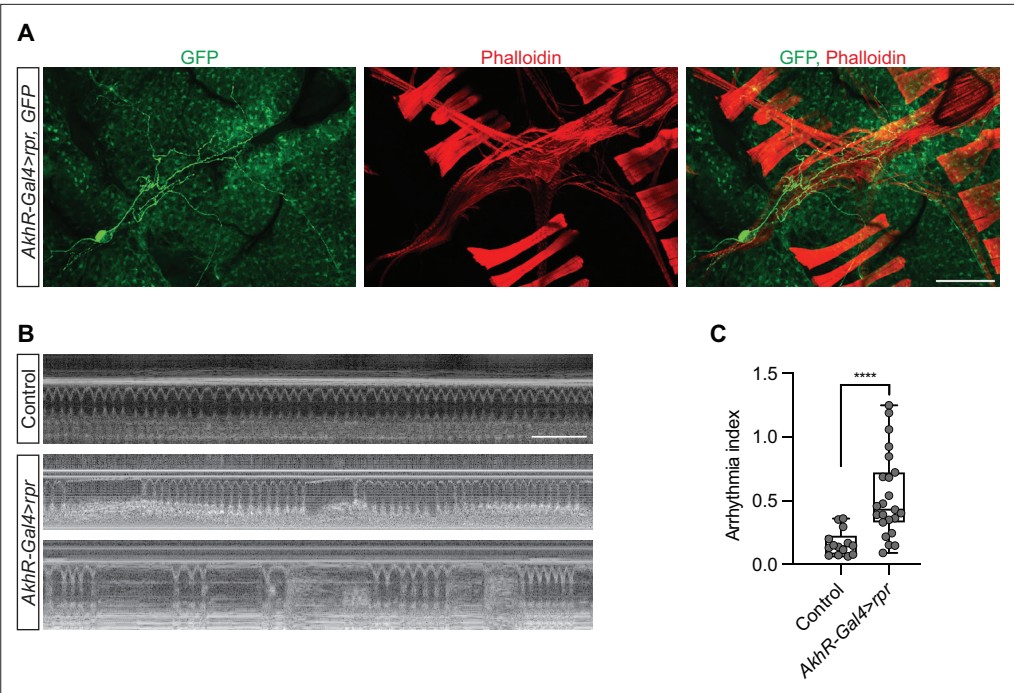

**Figure 6.** Partial elimination of ACNs causes arrhythmia. (**A**) A representative confocal image of an adult *AkhR*-Gal4>*rpr*, *GFP* (*AkhR*-Gal4>UAS-*rpr*, UAS-*GFP*) fly (7-day-old, female). AkhR >*GFP* labels AkhR in green. *AkhR, adipokinetic hormone receptor; rpr, reaper*. Phalloidin stains actin filaments red. Scale bar: 100 μm. (**B**) Representative images obtained from optical coherence tomography (OCT) videos of control (*w*[1118]) and *AkhR*-Gal4>*rpr* (*AkhR*-Gal4>UAS-*rpr*, UAS-*GFP*) flies (7-day-old, female). Scale bar: 2 seconds. (**B**) Quantitation of arrhythmia index in control (n=14 NFD) and *AkhR*-Gal4>*rpr* (n=23 NFD). Statistical analysis was performed using *t*-test corrected with Welch; ****p<0.0001. The numerical data used to generate the figure are provided in *Figure 6—source data 1*.

The online version of this article includes the following source data for figure 6:

**Source data 1.** The numerical data used to generate the *Figure 6*.

tissue, into glucose that is released into the blood stream. Its link to cardiac arrhythmia has been well-established. The glucose signaling pathway is conserved across species, including *Drosophila* (*Kim and Rulifson, 2004*). Like in humans, in flies, the glucagon-like hormone Akh regulates the glucose levels in hemolymph (fly equivalent to blood) (*Kim and Rulifson, 2004*). Under starvation conditions, Akh mobilizes glycogen and lipids to maintain the circulatory glucose levels. The flies become hyperactive and show increased food searching behavior. Flies without Akh, either by depletion or APC elimination, show starvation-resistant behavior, suggesting that the behavioral change is mediated by Akh (*Huang et al., 2020*; *Lee and Park, 2004*; *Yu et al., 2016*). Notably, in flies fed a HFD, the AkhR-expressing neurons in the brain become hypersensitive to Akh stimulation, due to upregulated AkhR (*Huang et al., 2020*). Finally, prepupae injected with low pharmacological doses of Akh, that is, doses higher than physiologically expected, showed increased heart rates (*Noyes et al., 1995*). Thus, in both humans and flies, heart rate and rhythm respond to nutritional changes (flies, see *Figure 1*), this mechanism likely supports the higher metabolic demands and possibly mediates the switch from glucose to stored lipids as the main energy source.

To date, studies have focused on the direct effect of glucagon on cardiac tissue, based on the notion that cardiomyocytes express glucagon receptors (GCGR). However, evidence of GCGR expression in human heart tissue has been conflicting. No cardiac signal was detected when using radioactively labeled glucagon (*Bomholt et al., 2022*), results of mRNA data have been inconsistent (*Aranda-Domene et al., 2023*; *Bomholt et al., 2022*), and protein data for GCGR to support cardiac expression is lacking (*Neumann et al., 2023*). Here, we identified an endocrine neuron-heart axis using a fly model for HFD-induced arrhythmia. In this fly model (*Figure 7*), the consumption of a HFD elevates the expression of Akh, a glucagon-like hormone, and activity of the APC. The elevated circulatory Akh

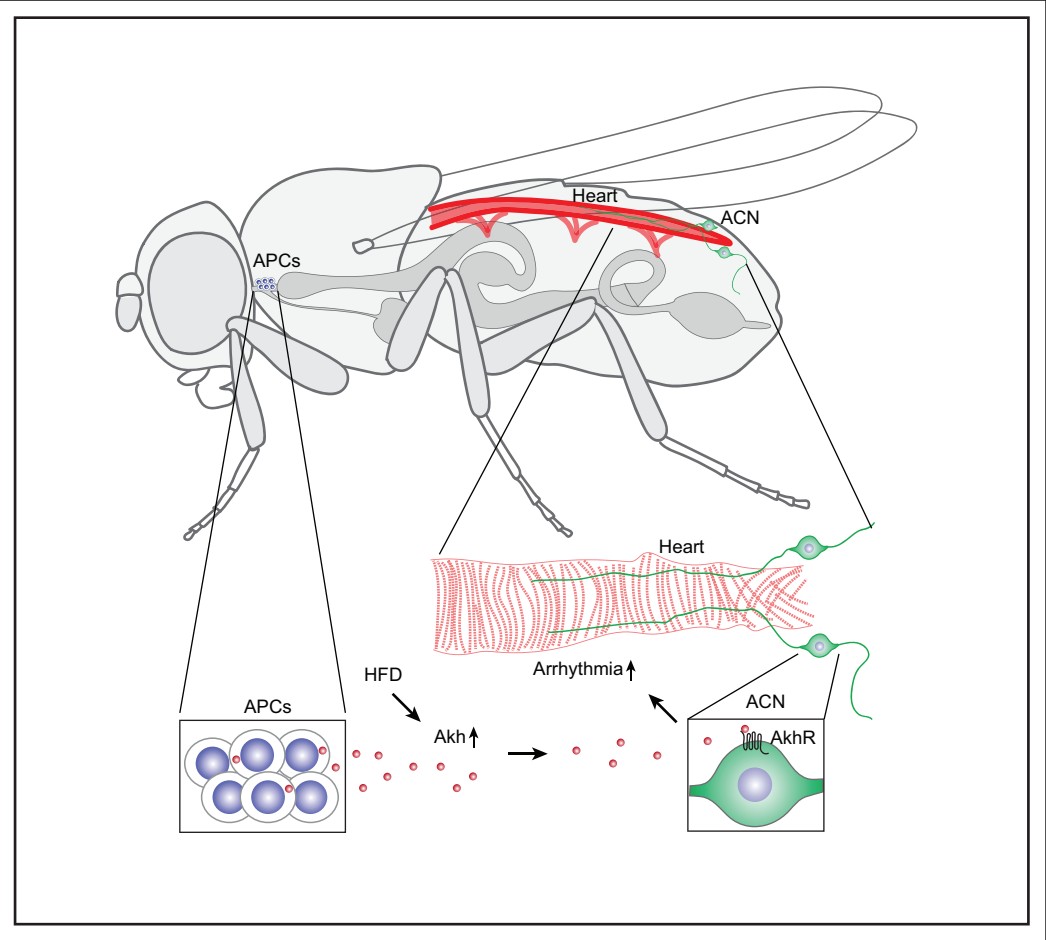

**Figure 7.** Model: Akh/AkhR mediates the high-fat diet (HFD)-induced cardiac arrhythmia. An HFD upregulates adipokinetic hormone (Akh) levels in the Akh-producing cells (APCs) located at the corpora cardiaca adjacent to the anterior esophagus. HFD also increases APC activity; this results in increased Akh secretion into the circulation. Akh acts on the Akh receptors (AkhR) expressed by the AkhR cardiac neurons (ACN). The ACN are a pair of neurons located at the posterior end of the adult fly heart that regulate the heartbeat. Through these signaling pathways, increased Akh under HFD conditions leads to increased arrhythmia.

is transported to the AkhR cardiac neurons (ACN) located near the posterior end of the heart, where the increased signaling leads to an escalated heart arrhythmia index. It is noteworthy that Akh injection induces cardioacceleration in prepupae (*Noyes et al., 1995*). These findings revealed a hitherto undescribed regulatory signaling pathway that links the consumption of superfluous macronutrients with heart arrhythmia.

## Cardiac regulatory neurons in *Drosophila*

*Drosophila* have a cardiac cycle that consists of alternating retrograde and anterograde heartbeats that correlate to the multi-chamber diastole and systole, respectively (*Dulcis and Levine, 2005a*). Accordingly, two signaling systems that innervate the fly heart were identified. The retrograde beat is being regulated by glutamatergic innervation. Its transverse nerves run bilaterally along the longitudinal muscle and innervate the cardiac muscles of the conical chamber as well as the alary muscles (*Dulcis and Levine, 2005a*; *Dulcis and Levine, 2003*). The anterograde heartbeat is being regulated by crustacean cardioactive peptide (CCAP) innervation. Its bipolar neurons (BpN) extend CCAP fibers that innervate each segment of the abdominal heart. Four additional large CCAP-positive neurons innervate the terminal chamber (*Dulcis et al., 2005b*; *Dulcis and Levine, 2003*). These earlier studies leave the possibility of synaptic input or hormonal regulation of the BpN neurosecretory signals to the heart. The authors comment on the absence of direct descending input from the corpora cardiaca to

the *Drosophila* abdominal heart as has been observed in other systems (*Dulcis and Levine, 2003*). Here, we identified corpora cardiaca released Akh signaling to previously unreported AkhR cardiac neurons (ACN) that could exercise this function in flies. Given their localization at the terminal end of the cardiac chamber, where the four large CCAP-positive neurons are located as well, it is possible that the ACN modulate CCAP signaling at the heart. In fact, the previous study detected synaptotagmin on the BpN (BpN6) cell bodies, which suggests the presence of presynaptic sites (*Dulcis and Levine, 2003*). We are currently investigating if the BpN and ACN act together, whether in concert or opposingly.

### Relevance of findings in flies for humans

This study revealed how glucagon-like hormone Akh is released by APCs in response to HFD and stimulates AkhR cardiac neurons (ACNs) to regulate heart rhythm in flies. Like in the flies, glucagon infusion in healthy human volunteers induces arrhythmias (*Jaca et al., 2002*; *Markiewicz et al., 1978*). Moreover, a preliminary study in infants and children demonstrated the potential of glucagon to treat atrioventricular (AV) block, a heart rhythm disorder marked by a slow heart rate caused by dysfunctional electrical conduction (*Hurwitz, 1973*). Clinical guidelines by the American Heart Association recommend the use of glucagon to treat bradycardia due to beta-blocker or calcium channel blocker overdose (*Kusumoto et al., 2019*). However, the mechanism by which glucagon exerts these beneficial clinical effects remains poorly understood. Our findings implicate a potentially conserved signaling pathway in which elevated glucagon leads to increased cardiac neuronal activity and subsequent increased heart rate and arrhythmia. Multiple trials are investigating the possible benefits of glucagon receptors (GCGR) agonists alone or in combination with GLP-1 agonists to treat arrhythmia, but so far the results have been mixed. An initial trial with a GCGR antagonist in patients with type 2 diabetes was halted prematurely due to the detrimental side effects, which included elevated blood cholesterol and steatosis (liver dysfunction) (*Agrawal and Gupta, 2016*; *Kazda et al., 2016*; *Pearson et al., 2016*). Our findings provide new possible glucagon-related targets for treating obesity-associated cardiac arrhythmia by targeting cardiac neurons receptive to glucagon signaling rather than the cardiomyocytes, as had been the leading hypothesis to date. Studies into the human equivalent of ACN would be interesting; possibly, the vagus nerve could fulfill a similar function as its importance for cardiac rhythm has been firmly established (*Ambache and Lippold, 1949*; *Cai et al., 2023*; *Freeman, 1951*; *González et al., 2023*; *Kharbanda et al., 2023*; *van Weperen and Vaseghi, 2023*).

Finally, while the exact implications of these findings for new therapeutics remain to be seen, they do support the avoidance of a HFD. If unavoidable, for example, in the case of a therapeutic ketogenic diet or in the subset of patients with diabetes who have elevated glucagon blood levels (*D'Alessio, 2011*; *Unger et al., 1970*), then targeting the glucagon pathway might protect against harmful cardiac side effects.

## Materials and methods

**Key resources table**

| Reagent type (species) or resource | Designation | Source or reference | Identifiers | Additional information |
|---|---|---|---|---|
| Antibody | Mouse monoclonal anti-Brp | Developmental Studies Hybridoma Bank | Cat# Nc82 RRID:AB_2314866 | IF (1:100) |
| Antibody | Goat polyclonal anti-mouse Alexa Fluor 647 | Jackson Immunoresearch Laboratories | Cat# 115-605-003 RRID:AB_2338902 | IF (1:500) |
| Antibody | Rabbit polyclonal anti-Akh | Prof. Jae Park | N.A. | IF (1:1000) |
| Antibody | Goat polyclonal anti-rabbit Alexa Fluor 568 | Thermo Fisher | Cat# A-11011 RRID:AB_143157 | IF (1:500) |
| Other | Alexa Fluor 647 Phalloidin | Thermo Fisher | Cat# A22287 | IF (1:1000) |
| Other | Normal goat serum | Jackson Immunoresearch Laboratories | Cat# 102643-594 | 1% in 1xPBST |

*Continued on next page*

*Continued*

| Reagent type (species) or resource | Designation | Source or reference | Identifiers | Additional information |
|---|---|---|---|---|
| Genetic reagent (*Drosophila melanogaster*) | *w*[1118] | Bloomington *Drosophila* Stock Center | RRID:BDSC_3605 | |
| Genetic reagent (*D. melanogaster*) | *Akh*-Gal4 | Bloomington *Drosophila* Stock Center | RRID:BDSC_25684 | |
| Genetic reagent (*D. melanogaster*) | UAS-*Akh*-RNAi | Bloomington *Drosophila* Stock Center | RRID:BDSC_34960 | |
| Genetic reagent (*D. melanogaster*) | *w*[1118] P{UAS-rpr.C}27; P{UAS-2x*EGFP*}AH3 | Bloomington *Drosophila* Stock Center | RRID:BDSC_ 91417 | |
| Genetic reagent (*D. melanogaster*) | *AkhR*[null] | Bloomington *Drosophila* Stock Center | RRID:BDSC_80937 | |
| Genetic reagent (*D. melanogaster*) | w[*]; P{w[+mC]=LexAop-CD8-GFP-2A-CD8-GFP}2; P{w[+mC]=UAS-*mLexA-VP16-NFAT*}H2, P{w[+mC]=lexAop-rCD2-GFP}3/TM6B, Tb[1] | Bloomington *Drosophila* Stock Center | RRID:BDSC_66542 | |
| Genetic reagent (*D. melanogaster*) | *10x*UAS-*GFP* | Bloomington *Drosophila* Stock Center | RRID:BDSC_32185 | |
| Genetic reagent (*D. melanogaster*) | *AkhR-T2A*-Gal4 | Bloomington *Drosophila* Stock Center | RRID:BDSC_78877 | |
| Chemical compound, drug | 4% paraformaldehyde (PFA) | Thermo Fisher Scientific | Cat# J19943.K2 | |
| Chemical compound, drug | Triton X-100 | Sigma-Aldrich | Cat# 9002-93-1 | 0.2% in 1xPBS |
| Chemical compound, drug | 4′,6-Diamidino-2-phenylindole (DAPI) | Thermo Fisher Scientific | Cat# D1306 RRID:AB_2629482 | (0.5 µg/ml) |
| Commercial assay or kit | 10x phosphate buffered saline | Quality Biological | Cat# 119-069-131 | |
| Commercial assay or kit | TRIzol | Thermo Fisher Scientific | Cat# 15596026 | |
| Commercial assay or kit | EcoDry Premix kit | Takara Bio | Cat# 639549 | |
| Commercial assay or kit | PowerSYBR Green | Applied Biosystems, Thermo Fisher Scientific | Cat# 4367659 | |
| Commercial assay or kit | VECTASHIELD | Vector Laboratories | Cat# H-1000 | |
| Software, algorithm | Adobe Illustrator 2022 | Adobe Inc. | RRID:SCR_010279 | |
| Software, algorithm | ImageJ | National Institutes of Health | RRID:SCR_003070 | |
| Software, algorithm | GraphPad Prism9 | GraphPad | RRID:SCR_002798 | |

## *Drosophila* husbandry

*Drosophila* were reared on a NFD (Nutri-Fly German formula; Genesee Scientific, San Diego, CA; 66-115) or a HFD (NFD supplemented with 14% coconut oil), under standard conditions (25°C, 60% humid, 12 hour:12 hour dark:light). The following fly lines were obtained from Bloomington Drosophila Stock Center (BDSC) at Indiana University Bloomington (Bloomington, IN): *w*[1118] (BDSC_3605), *Akh*-Gal4 (BDSC_25684), UAS-*Akh*-RNAi (BDSC_34960), *Akh-T2A*-Gal4 (BDSC_78877), *w*[1118] P{UAS-*rpr.C*}27; P{UAS-2x*EGFP*}AH3 (BDSC_91417), *AkhR*[null] (BDSC_80937), w[*]; P{w[+mC]=LexAop-CD8-GFP-2A-CD8-GFP}2; P{w[+mC]=UAS-*mLexA-VP16-NFAT*}H2, P{w[+mC]=lexAop-rCD2-GFP}3/TM6B, Tb[1] (BDSC_66542), and *10x*UAS-*GFP* (BDSC_32185).

## Optical coherence tomography (OCT)

Cardiac function in adult *Drosophila* was measured using an OCT system (Bioptigen; this system was built by the Biophotonics Group, Duke University, Durham, NC; *Yelbuz et al., 2002*). For this, newly eclosed adult flies (female) were transferred to fresh vials containing NFD or HFD for 7 days. The flies were anesthetized by carbon dioxide ($CO_2$) and mounted onto a glass slide using fixogum rubber cement (Marabu GmbH & Co. KG, Tamm, Germany; MR290117000). The flies were allowed to recover for 10 minutes and then imaged using OCT. The flies were scanned at the cardiac chamber in abdominal segment A2. The acquisition parameters were as follows: 48 Hz, 800 frames.

## Quantification of heart period and arrhythmia index

The heart period and arrhythmia index were quantified as previously reported (*Ocorr et al., 2007*) with minor modifications. In brief, the OCT raw data was processed using ImageJ software (version 2.9.0/1.53) (*Schneider et al., 2012*). Within each OCT image (10 seconds), the heart periods were arranged by length. The heart period value for each fly was based on the average of three short, three medium, and three long heart periods (i.e., nine periods in total). The arrhythmia index for a genotype was based on the group average (7-day-old females; n=11–23), presented with the standard deviation (SD).

## Immunostaining and confocal imaging

Flies were dissected in 1xPBS, then fixed in 4% PFA for 30 minutes at room temperature. The specimens were then washed in PBST (0.2% Triton X-100 in 1xPBS) three times, 15 minutes each, followed by blocking in 1% normal goat serum (Jackson ImmunoResearch Laboratories, West Grove, PA; 102643–594) in PBST for 1 hour at room temperature. The specimens were incubated in primary antibody at 4°C overnight, then washed in PBST three times for 15 minutes. Next, the specimens were incubated in secondary antibody for 2 hours at room temperature, followed by three 15-minute washes in PBST, followed by a 10-minute wash in PBS. DAPI staining (Thermo Fisher Scientific, Waltham, MA; D1306, 0.5 µg/ml in PBST) was performed in-between the washing steps after the secondary antibody staining. The following antibodies were used: mouse monoclonal anti-Brp 1:100 (Developmental Studies Hybridoma Bank, Iowa City, IA; nc82, RRID:AB_2314866); rabbit anti-Akh (1:1000); goat anti-rabbit Alexa Fluor 568 1:500 (Thermo Fisher; A-11011); goat anti-mouse Alexa Fluor 647 1:500 (Jackson ImmunoResearch Laboratories; 115-605-003). Alexa Fluor 647 Phalloidin 1:1000 (Thermo Fisher Scientific; A22287) was used to stain the filament actin. Antibodies were diluted in the blocking buffer. The specimens were mounted with Vectashield antifade mounting media (Vector Laboratories, Newark, CA; H-1000). The samples were imaged using a ZEISS LSM 900 confocal microscope and ZEISS ZEN blue edition (version 3.0) acquisition software under a 20× Plan-Apochromat 0.8 N.A. air objective or a 63× Plan-Apochromat 1.4 N.A. oil objective. For quantitative comparison of intensities, settings were chosen to avoid oversaturation (using range indicator in ZEN blue) and applied across images for all samples within an assay. ImageJ was used for image processing (version 2.9.0/1.53t; National Institutes of Health, Bethesda, MD) (*Schneider et al., 2012*).

## Reverse transcriptase-quantitative PCR (RT-qPCR)

For quantification of Akh and AkhR mRNA levels, adult female flies fed NFD or HFD for 7 days were transferred into 1.5 ml Eppendorf tubes (10 flies/tube) and flash frozen in liquid nitrogen. Total RNA was extracted using TRIzol (Thermo Fisher Scientific, Cat# 15596026) according to the manufacturer's protocol. In brief, 0.5 ml TRIzol was added to each tube, then flies were grinded using a pestle. Following, 0.1 ml chloroform was added, sample tubes securely capped, and vigorously shaken by hand for 15 seconds. Samples were incubated at 22°C for 2 minutes, then centrifuged at 12,000 × *g* for 15 minutes at 4°C, and supernatants transferred to a clean tube in which total RNA was precipitated using isopropanol (Sigma-Aldrich, St. Louis, MO; 67-63-0). Synthesis of cDNA using reverse transcriptase was performed with the RNA to cDNA EcoDry Premix kit (Takara Bio, San Jose, CA; 639549) and the subsequent quantitative PCR using PowerSYBR Green (Applied Biosystems, Thermo Fisher Scientific; 4367659), according to the manufacturers' protocols. qPCR was performed using a QuantStudio 6 Pro Real-Time PCR machine (Thermo Fisher Scientific). The following primers (5'–3') were used: Akh_F TTTCGAGACACAGCAGGGCA, with Akh_R GGTGCTTGCAGTCCAGAAA; AkhR_F ACAATCCGTCGGTGAACA, with AkhR_R CATCACCTGGCCTCTTCCAT. For each treatment, three biological replicates were prepared. ΔΔCT method was used to quantify the gene expression levels with normalization to *Ribosomal protein L32* (*RpL32*) as internal reference gene (Rpl32_F AACCGCGT TTACTGCGGCGA, with Rpl32_R AGAACGCAGGCGACCGTTGG).

## CaLexA assays

CaLexA assays were performed to determine the neuron activity. *Akh*-Gal4 virgins were crossed with CaLexA males (*w[*]*; *P{w[+mC]=LexAop-CD8-GFP-2A-CD8-GFP}2*; *P{w[+mC]=UAS-mLexA-VP16-NFAT}H2*, *P{w[+mC]=lexAop-rCD2-GFP}3/TM6B, Tb[1]*) and maintained on NFD. The newly eclosed adults were transferred to fresh food vials containing NFD or HFD for 3 days. The flies were then

dissected in 1x phosphate buffered saline (1xPBS) (Quality Biological, Gaithersburg, MD; 119-069-131) at room temperature and fixed in 4% paraformaldehyde (PFA) (Thermo Fisher Scientific; J19943. K2) in 1xPBS for 1 hour at room temperature, followed by three 15-minute washes in 1xPBS with Triton X-100 (Sigma-Aldrich; 9002-93-1) (PBST). The specimens were then stained in DAPI (0.5 µg/ml in PBST) (Thermo Fisher Scientific; D1306) for 10 minutes and washed once for 15 minutes in 1xPBS before mounting with Vectashield antifade mounting media (Vector Laboratories; H-1000). The samples were imaged using a ZEISS LSM 900 confocal microscope and ZEISS ZEN blue edition (version 3.0) acquisition software under a 20× Plan-Apochromat 0.8 N.A. air objective and a 63× Plan-Apochromat 1.4 N.A. oil objective. For quantitative comparison of intensities, settings were chosen to avoid oversaturation (using range indicator in ZEN blue) and applied across images for all samples within an assay. ImageJ was used for image processing (version 2.9.0/1.53t; National Institutes of Health) (*Schneider et al., 2012*).

## Data analysis and figure preparation

Figures were arranged using Adobe Illustrator software (version 26.2.1; Adobe Inc, San Jose, CA). The relative fluorescence intensity was acquired using ImageJ software (version 2.9.0/1.53t; National Institutes of Health) (*Schneider et al., 2012*). Data plotting and statistical analyses were performed using Prism 9 (GraphPad Software, Boston, MA). For box plots, midlines represent the median, and whiskers indicate the minimum and maximum values. Data normality was tested by using the Shapiro–Wilk test. Normally distributed data were analyzed by Student's *t*-test with Welch's correction (two groups) or by a one-way ANOVA followed by Dunnett's correction or two-way ANOVA corrected with Sidak. p-value <0.05 was considered significant.

## Acknowledgements

We would like to thank the Bloomington Drosophila Stock Center (BDSC) based at Indiana University Bloomington (Bloomington, IN) for the Drosophila stocks, Prof. Jae Park (the University of Tennessee) for providing the antibodies, and the Developmental Studies Hybridoma Bank (DSHB) based at the University of Iowa (Iowa City, IA) for providing the antibodies. This work was supported by National Institutes of Health grants NHLBI R01-HL134940 (ZH), R01-HL180768 (ZH), and NICHD R01-HD111480 (ZH).

## Additional information

### Funding

| Funder | Grant reference number | Author |
| --- | --- | --- |
| National Heart Lung and Blood Institute | R01-HL180768 | Zhe Han |
| Eunice Kennedy Shriver National Institute of Child Health and Human Development | R01-HD111480 | Zhe Han |
| National Heart Lung and Blood Institute | R01-HL134940 | Zhe Han |

The funders had no role in study design, data collection and interpretation, or the decision to submit the work for publication.

### Author contributions

Yunpo Zhao, Conceptualization, Formal analysis, Investigation, Visualization, Methodology, Writing – original draft, Writing – review and editing; Jianli Duan, Conceptualization, Formal analysis, Investigation, Methodology; Joyce van de Leemput, Writing – original draft, Writing – review and editing; Zhe Han, Conceptualization, Supervision, Funding acquisition, Writing – original draft, Writing – review and editing

## Author ORCIDs
Yunpo Zhao http://orcid.org/0000-0002-7942-3406
Zhe Han https://orcid.org/0000-0002-5177-7798

Reviewer #1 (Public review): https://doi.org/10.7554/eLife.94512.4.sa1
Reviewer #3 (Public review): https://doi.org/10.7554/eLife.94512.4.sa2
Author response https://doi.org/10.7554/eLife.94512.4.sa3

## Additional files

### Supplementary files
MDAR checklist

### Data availability
All relevant data can be found within the article and its supplementary information.

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
