## [Editor Report · eLife Assessment]

This study reports **useful** information on the mechanisms by which a high-fat diet induces arrhythmias in the model organism *Drosophila*. Specifically, the authors propose that adipokinetic hormone (Akh) secretion is increased with this diet, and through binding of Akh to its receptor on cardiac neurons, arrhythmia is induced. The authors have revised their manuscript, but in some areas, the evidence remains **incomplete**, which the authors say future studies will be directed to closing the present gaps. Nonetheless, the data presented will be helpful to those who wish to extend the research to a more complex model system, such as the mouse.

---

## [Referee Report · Reviewer #1 (Public review)]

Summary:

In the manuscript submission by Zhao et al. entitled, "Cardiac neurons expressing a glucagon-like receptor mediate cardiac arrhythmia induced by high-fat diet in Drosophila" the authors assert that cardiac arrhythmias in Drosophila on a high fat diet is due in part to adipokinetic hormone (Akh) signaling activation. High fat diet induces Akh secretion from activated endocrine neurons, which activate AkhR in posterior cardiac neurons. Silencing or deletion of Akh or AkhR blocks arrhythmia in Drosophila on high fat diet. Elimination of one of two AkhR expressing cardiac neurons results in arrhythmia similar to high fat diet.

Strengths:

The authors propose a novel mechanism for high fat diet induced arrhythmia utilizing the Akh signaling pathway that signals to cardiac neurons.

---

## [Referee Report · Reviewer #3 (Public review)]

Zhao et al. provide new insights into the mechanism by which a high-fat diet (HFD) induces cardiac arrhythmia employing Drosophila as a model. HFD induces cardiac arrhythmia in both mammals and Drosophila. Both glucagon and its functional equivalent in Drosophila Akh are known to induce arrhythmia. The study demonstrates that Akh mRNA levels are increased by HFD and both Akh and its receptor are necessary for high-fat diet-induced cardiac arrhythmia, elucidating a novel link. Notably, Zhao et al. identify a pair of AKH receptor-expressing neurons located at the posterior of the heart tube. Interestingly, these neurons innervate the heart muscle and form synaptic connections, implying their roles in controlling the heart muscle. The study presented by Zhao et al. is intriguing, and the rigorous characterization of the AKH receptor-expressing neurons would significantly enhance our understanding of the molecular mechanism underlying HFD-induced cardiac arrhythmia.

Many experiments presented in the manuscript are appropriate for supporting the conclusions while additional controls and precise quantifications should help strengthen the authors' arguments. The key results obtained by loss of Akh (or AkhR) and genetic elimination of the identified AkhR-expressing cardiac neurons do not reconcile, complicating the overall interpretation.

The most exciting result is the identification of AkhR-expressing neurons located at the posterior part of the heart tube (ACNs). The authors attempted to determine the function of ACNs by expressing rpr with AkhR-GAL4, which would induce cell death in all AkhR-expressing cells, including ACNs. The experiments presented in Figure 6 are not straightforward to interpret. Moreover, the conclusion contradicts the main hypothesis that elevated Akh is the basis of HFD-induced arrhythmia. The results suggest the importance of AkhR-expressing cells for normal heartbeat. However, elimination of Akh or AkhR restores normal rhythm in HFD-fed animals, suggesting that Akh and AkhR are not important for maintaining normal rhythms. If Akh signaling in ACNs is key for HFD-induced arrhythmia, genetic elimination of ACNs should unalter rhythm and rescue the HFD-induced arrhythmia. An important caveat is that the experiments do not test the specific role of ACNs. ACNs should be just a small part of the cells expressing AkhR. Specific manipulation of ACNs will significantly improve the study. Moreover, the main hypothesis suggests that HFD may alter the activity of ACNs in a manner dependent on Akh and AkhR. Testing how HFD changes calcium, possibly by CaLexA (Figure 2) and/or GCaMP, in wild-type and AkhR mutant could be a way to connect ACNs to HFD-induced arrhythmia. Moreover, optogenetic manipulation of ACNs may allow for specific manipulation of ACNs.

Interestingly, expressing rpr with AkhR-GAL4 was insufficient to eliminate both ACNs. It is not clear why it didn't eliminate both ACNs. Given the incomplete penetrance, appropriate quantifications should be helpful. Additionally, the impact on other AhkR-expressing cells should be assessed. Adding more copies of UAS-rpr, AkhR-GAL4, or both may eliminate all ACNs and other AkhR-expressing cells. The authors could also try UAS-hid instead of UAS-rpr.

---

## [Author Response]

The following is the authors’ response to the previous reviews

**Public Reviews:**

**Reviewer #1 (Public review):**
Summary:In the manuscript submission by Zhao et al. entitled, "Cardiac neurons expressing a glucagon-like receptor mediate cardiac arrhythmia induced by high-fat diet in Drosophila" the authors assert that cardiac arrhythmias in Drosophila on a high fat diet is due in part to adipokinetic hormone (Akh) signaling activation. High fat diet induces Akh secretion from activated endocrine neurons, which activate AkhR in posterior cardiac neurons. Silencing or deletion of Akh or AkhR blocks arrhythmia in Drosophila on high fat diet. Elimination of one of two AkhR expressing cardiac neurons results in arrhythmia similar to high fat diet.Strengths:The authors propose a novel mechanism for high fat diet induced arrhythmia utilizing the Akh signaling pathway that signals to cardiac neurons.Comments on revisions:The authors have addressed my other concerns. The only outstanding issue is in regard to the following comment:The authors state that "HFD led to increased heartbeat and an irregular rhythm." In representative examples shown, HFD resulted in pauses, slower heart rate, and increased irregularity in rhythm but not consistently increased heart rate (Figures 1B, 3A, and 4C). Based on the cited work by Ocorr et al (https://doi.org/10.1073/pnas.0609278104), Drosophila heart rate is highly variable with periods of fast and slow rates, which the authors attributed to neuronal and hormonal inputs. Ocorr et al then describe the use of "semi-intact" flies to remove autonomic input to normalize heart rate. Were semi-intact flies used? If not, how was heart rate variability controlled? And how was heart rate "increase" quantified in high fat diet compared to normal fat diet? Lastly, how does one measure "arrhythmia" when there is so much heart rate variability in normal intact flies?The authors state that 8 sec time windows were selected at the discretion of the imager for analysis. I don't know how to avoid bias unless the person acquiring the imaging is blinded to the condition and the analysis is also done blind. Can you comment whether data acquisition and analysis was done in a blinded fashion? If not, this should be stated as a limitation of the study.

Drosophila heart rate is highly variable. During the recording, we were biased to choose a time window when heartbeat was fairly stable. This is a limitation of the study, which we mentioned in the revised version. We chose to use intact over “semi-intact” flies with an intention to avoid damaging the cardiac neurons.

**Reviewer #3 (Public review):**
Zhao et al. provide new insights into the mechanism by which a high-fat diet (HFD) induces cardiac arrhythmia employing Drosophila as a model. HFD induces cardiac arrhythmia in both mammals and Drosophila. Both glucagon and its functional equivalent in Drosophila Akh are known to induce arrhythmia. The study demonstrates that Akh mRNA levels are increased by HFD and both Akh and its receptor are necessary for high-fat diet-induced cardiac arrhythmia, elucidating a novel link. Notably, Zhao et al. identify a pair of AKH receptor-expressing neurons located at the posterior of the heart tube. Interestingly, these neurons innervate the heart muscle and form synaptic connections, implying their roles in controlling the heart muscle. The study presented by Zhao et al. is intriguing, and the rigorous characterization of the AKH receptor-expressing neurons would significantly enhance our understanding of the molecular mechanism underlying HFD-induced cardiac arrhythmia.Many experiments presented in the manuscript are appropriate for supporting the conclusions while additional controls and precise quantifications should help strengthen the authors' arguments. The key results obtained by loss of Akh (or AkhR) and genetic elimination of the identified AkhR-expressing cardiac neurons do not reconcile, complicating the overall interpretation.

We thank the reviewer for the positive comments. We believe that more signaling pathways are active in the AkhR neurons and regulate rhythmic heartbeat. We are current searching for the molecules and pathways that act on the AkhR cardiac neurons to regulate the heartbeat. Thus, AkhR neuron x shall have a more profound effect. Loss of AkhR is not equivalent to AkhR neuron ablation.

The most exciting result is the identification of AkhR-expressing neurons located at the posterior part of the heart tube (ACNs). The authors attempted to determine the function of ACNs by expressing rpr with AkhR-GAL4, which would induce cell death in all AkhRexpressing cells, including ACNs. The experiments presented in Figure 6 are not straightforward to interpret. Moreover, the conclusion contradicts the main hypothesis that elevated Akh is the basis of HFD-induced arrhythmia. The results suggest the importance of AkhR-expressing cells for normal heartbeat. However, elimination of Akh or AkhR restores normal rhythm in HFD-fed animals, suggesting that Akh and AkhR are not important for maintaining normal rhythms. If Akh signaling in ACNs is key for HFD-induced arrhythmia, genetic elimination of ACNs should unalter rhythm and rescue the HFD-induced arrhythmia. An important caveat is that the experiments do not test the specific role of ACNs. ACNs should be just a small part of the cells expressing AkhR. Specific manipulation of ACNs will significantly improve the study. Moreover, the main hypothesis suggests that HFD may alter the activity of ACNs in a manner dependent on Akh and AkhR. Testing how HFD changes calcium, possibly by CaLexA (Figure 2) and/or GCaMP, in wild-type and AkhR mutant could be a way to connect ACNs to HFD-induced arrhythmia. Moreover, optogenetic manipulation of ACNs may allow for specific manipulation of ACNs.

We thank the reviewer for suggesting the detailed experiments and we believe that address these points shall consolidate the results. As AkhR-Gal4 also expresses in the fat body, we set out to build a more specific driver. We planned to use split-Gal4 system (Luan et al. 2006. PMID: 17088209). The combination of pan neuronal Elav-Gal4.DBD and AkhRp65.AD shall yield AkhR neuron specific driver. We selected 2580 bp AkhR upstream DNA and cloned into pBPp65ADZpUw plasmid (Addgene plasmid: #26234). After two rounds of injection, however, we were not able to recover a transgenic line.

We used GCaMP to record the calcium signal in the AkhR neurons. AkhR-Gal4>GCaMP has extremely high levels of fluorescence in the cardiac neurons under normal condition.

We are screening Gal4 drivers, trying to find one line that is specific to the cardiac neurons and has a lower level of driver activity.

Interestingly, expressing rpr with AkhR-GAL4 was insufficient to eliminate both ACNs. It is not clear why it didn't eliminate both ACNs. Given the incomplete penetrance, appropriate quantifications should be helpful. Additionally, the impact on other AhkR-expressing cells should be assessed. Adding more copies of UAS-rpr, AkhR-GAL4, or both may eliminate all ACNs and other AkhR-expressing cells. The authors could also try UAS-hid instead of UASrpr.

We quantified the AkhR neuron ablation and found that about 69% (n=28) showed a single ACN in AkhR-Gal4>rpr flies. It is more challenging to quantify other AkhR-expressing cells, as they are wide-spread distributed. We tried to add more copies of UAS-rpr or AkhR-Gal4, which caused developmental defects (pupa lethality). Thus, as mentioned above, we are trying to find a more specific driver for targeting the cardiac neurons.

**Recommendations for the authors:**

**Reviewer #3 (Recommendations for the authors):**
The authors refer 'crop' as the functional equivalent of the human stomach. Considering the difference in their primary functions, this cannot be justified.

In Drosophila, the crop functions analogously to the stomach in vertebrates. It is a foregut storage and preliminary processing organ that regulates food passage into the midgut. It’s more than a simple reservoir. Crop engages in enzymatic mixing, neural control, and active motility.

Line 163 and 166, APCs are not neurons.

Akh-producing cells (APCs) in Drosophila are neuroendocrine cells, residing in the corpora cardiaca (CC). While they produce and secrete the hormone AKH (akin to glucagon), they are not brain interneurons per se. APCs share many neuronal features (vesicular release, axon-like projections) and receive neural inputs, effectively functioning as a peripheral endocrine center.